# Isometric Knee Muscle Strength and Patient-Reported Measures Five Years after Anterior Cruciate Ligament Reconstruction: Comparison of Single versus Dual Autograft Hamstring Tendon Harvesting

**DOI:** 10.3390/jcm11195682

**Published:** 2022-09-26

**Authors:** Ignacio Manchado, Luci M. Motta, Gustavo Blanco, Jesús González, Gerardo L. Garcés

**Affiliations:** 1Hospital Perpetuo Socorro, 35007 Las Palmas, Spain; 2Departamento de Ciencias Médicas y Quirúrgicas, University of Las Palmas de Gran Canaria, Paseo Blas Cabrera Felipe s/n, 35016 Las Palmas, Spain; 3Unidad de Investigación, Hospital Dr Negrin, 35007 Las Palmas, Spain

**Keywords:** anterior cruciate ligament, hamstring tendon graft, knee strength, IKDC, Lysholm

## Abstract

There is some controversy regarding the use of one or two hamstring tendons for anterior cruciate ligament reconstruction (ACLR). In this study, two cohorts of 22 male patients underwent an ACLR with hamstring tendon autografts. One cohort was reconstructed through an all-inside technique with the semitendinosus tendon (ST group) and the other with the semitendinosus and gracilis tendons (ST-G group). Anterior tibial translation (ATT), Lysholm, and IKDC scores were assessed preoperatively and five years postoperation. Additionally, isometric knee muscle strength was manually measured in both groups and in another cohort of 22 uninjured control male subjects five years after the operation. There were no significant differences in ATT and Lysholm scores between the operated groups. The IKDC score was lower in the ST-G group than in the ST group—9.57 (CI 14.89–4.25) (*p* < 0.001). No significant differences between injured and uninjured knees were detected in hamstring to quadriceps ratio strength and quadriceps limb symmetry index of the two operated groups, but the hamstring limb symmetry index was significantly lower in the ST-G group than in the ST and control groups. This study shows that using an ST-G autograft for ACLR yielded less flexor strength and worse results in some patient-reported outcome measures (PROM) than using an ST autograft five years after the operation. The observed results let us suggest that the use of one autograft hamstring tendon for ACLR is clinically preferable to the use of two hamstring tendons.

## 1. Introduction

In recent years, there has been an increase in hamstring graft preference over the “gold standard” bone–patellar tendon–bone (BPTB) for anterior cruciate ligament reconstruction (ACLR). Harvesting hamstrings has lower morbidity and reduces the risk of knee pain, kneeling pain, flexion limitation, and osteoarthritis [1,2]. Hamstring autografts most used are doubled semitendinosus and doubled gracilis tendons (ST-G), with a fixed or adjustable femoral device and an interference screw within the tibial tunnel [3,4,5,6,7]. This type of reconstruction is mostly carried out through a complete tibial tunnel to deliver the graft into the femoral socket. Another alternative is using a single bundle four-strand semitendinosus tendon graft (ST4), fixed to a femoral and tibial socket through an “all inside technique” (AIT) [3,4,7,8,9,10,11,12,13]. The potential advantages of AIT include preservation of bone stock, reduced incidence of complications (tibial plateau fractures, decreased cortical bone periosteal disruption, decreased postoperative pain), improved bone–graft integration, accelerated graft maturation, increased precision of anatomic placement [11,14,15,16], and, usually, an ST4 graft harvest with a larger diameter graft [5,7,17].

To assess successful results after an ACLR, it is necessary to evaluate knee laxity, limb strength, and patient-reported outcome measures (PROMs) [18]. Regarding laxity, it seems that there are no differences between all-inside techniques and full tibial tunnel techniques using hamstring tendons [3,4,10,11,12,13]. Considering self-reported knee function, most studies report that there are no differences in the results between the use of one or two hamstring tendons [3,4,8,10,11,13], but other authors [12] have noted that there may be minor nonsignificant differences.

Isolated ST tendon harvesting for an all-inside ACLR may result in fewer strength deficits than harvesting both ST-G hamstrings for the complete tibial tunnel technique [3,8,10,19]. However, this issue is not clear. Some authors point out that there are no differences in knee muscle strength between the reconstructed and their contralateral knees after using one or two hamstring autografts [4]; others refer that there are significant but not clinically relevant differences [3] and that differences depend on the knee angle during strength testing [10] or the angular velocity used during testing [12]. Nevertheless, most of the articles comparing the clinical results using the ST vs. ST-G hamstring grafts present data with a maximum follow-up of 3 years [3,7,10,12,19]. It is not known if the results could be different if assessed at a longer follow-up

After an ACLR, leg muscle strength is related to return to activity (RTA) or sport (RTS) [18,20,21,22], functional performance [23,24,25,26,27], and the risk of re-rupture [28,29]. Additionally, thigh muscle strength is correlated with self-reported patient outcomes [20,23,25,30,31,32,33,34,35,36,37]. The Limb Symmetry Index (LSI), for extensors and flexors, is the ratio of strength between the injured and uninjured knees. It is a common method of assessing the strength and functional performance after ACLR [26,35,38]. Another common method to assess progression after ACL injury is the hamstring/quadriceps (H/Q) ratio [39]. The H/Q ratio can be used to detect muscle imbalance, monitor knee joint stability, and indicate lower extremity injury prevention and rehabilitation [40,41,42]. It has been suggested that the H/Q ratio is more important than the maximal torque in the assessment of muscle function [40,41]. The H/Q ratio is altered in ACL-deficient and ACL-reconstructed knees compared with the uninjured contralateral limb and controls [39,41]. A decrease in relative hamstring strength combined with high relative quadriceps strength may present a potential risk factor for ACL tears [40,43,44,45].

There is controversy about using one versus two hamstring autografts for ACLR since this could influence postoperation knee strength [3,10,12,19]. Because of its importance in sports performance, knee muscle strength after ACLR has become an important parameter to assess objective results in the last few years. The most extended published follow-up was less than three years [3,7,10,12,19]. Knee strength, PROM, and anterior tibial translation (ATT) are important parameters for assessing results after ACLR. Our work aimed to study if, five years after the operation, the differential muscle strength between the reconstructed and its contralateral uninjured knee is different when the ST versus ST-G are harvested. We also aimed to study if there are differences in PROMs results and side-to-side ATT between the ST and ST-G groups five years after the operation. We hypothesized that no significant differences in the results would be detected and that the use of one of two tendons is equally safe and effective.

## 2. Materials and Methods

This is a retrospective observational study of two cohorts of 26 male patients who underwent unilateral ACLR 5 years previously and another cohort of uninjured male control patients. The study protocol was approved by the Human Research Ethics Committee at the University of Las Palmas de Gran Canaria (protocol number CEIH-2017-11), and the study was conducted in accordance with the principles of the 1975 Declaration of Helsinki. Informed consent to participate in the study was obtained from patients and controls. Informed consent was also obtained from one of their parents for each of the five underaged patients in the study.

### 2.1. Participants

Fifty-two male patients with symptomatic unilateral ACL deficiency, who underwent ACLR at the same hospital by the same surgical team in 2013, participated in the study. Inclusion criteria were unilateral ACL rupture diagnosed by the Lachman and Pivot–Shift positive test and side-to-side ATT differences >3 mm measured by a KT 1000TM (MedMetric: San Diego, CA, USA). The diagnosis was confirmed by magnetic resonance imaging (MRI). The patients were scheduled to undergo surgery due to symptomatic instability or patient–doctor decisions after a minimum of 4 weeks of rehabilitation. Exclusion criteria included a history of previous knee surgery or fracture around the knee, concomitancy of other knee ligament ruptures, and osteoarthritis greater than grade 2 (Kellgren and Lawrence classification). Concurrent meniscus tears or small chondral lesions were not criteria for patient exclusion. To be included in the study, patients needed to be involved in recreational sports activities 5 years postoperation.

Twenty-six patients underwent ACLR with a single bundle semitendinosus four-strand (ST group) arthroscopic all-inside reconstruction. The remaining 26 patients underwent arthroscopic ACLR with a single bundle combined doubled semitendinosus and double gracilis graft through a complete tibial tunnel (ST-G group). The mean age of the ST group was 26.8, and that of the ST-G group was 25.8. The third group of 22 uninjured male controls was recruited from a fitness center near the hospital five years after starting the study (Figure 1). Criteria for inclusion in this control group were lack of the previous injury in one or both lower limbs and participation in sports activities at a nonprofessional level. Controls were paired with patients for age and type of sports activity at follow-up revision, 5 years postoperation. Demographic data of the study population are presented in Table 1.

### 2.2. Surgical Methods

Operations were carried out arthroscopically by the same surgical team. Once ACL rupture was confirmed, an anteromedial longitudinal incision was made to obtain the graft from the hamstring tendons. Patients in the ST group underwent surgery with an all-inside technique. The semitendinosus tendon was harvested and prepared for the tape locking screw surgical technique (TLS) (Laboratoire FH: Mulhouse, France), following the single bundle four-strand semitendinosus graft technique described by Colette and Cassard [14]. The graft length was 50–55 mm, and the graft diameter was 9–10 mm.

In patients in the ST-G group, both the semitendinosus and gracilis tendons were harvested and prepared to create a single bundle combined doubled semitendinosus doubled gracilis graft. Through standard arthroscopic ACLR, femoral cortical fixation was achieved using XO Button (ConMed Linvatec: Largo, FL, USA). Tibial fixation was performed with a bioabsorbable interference screw (Matryx, ConMed Linvatec: Largo, FL, USA) in conjunction with an additional cortical staple over the out-of-the-tunnel portion of the graft.

The postoperative rehabilitation protocol was very similar for all patients in both groups, avoiding quadriceps contraction against gravity during the first 6 weeks after the operation. Running was allowed 3–4 months after the operation, and returning to sports activities was allowed 8–10 months post-reconstruction.

Patients were evaluated within 48 h prior to surgical reconstruction and 5 years postoperation. The control subjects were evaluated only once, coinciding with the fifth postoperative year of the patients. Patient-reported outcome measures (PROMs) were evaluated with the International Knee Documentation Committee (IKDC) subjective score and the Lysholm knee scoring scale. ATT was assessed using a KT-1000 arthrometer (MEDmetric: San Diego, CA, USA) through maximal manual traction with the knee at 30° of flexion. All tests were performed by the same two researchers independently (LM, GB) three times on both knees of the patients and controls, first on the uninjured knee and then on the injured knee. The mean value of the six measurements was used for the statistical analysis. Measures of the three tests carried out by the same observer and the six tests carried out by both coincided in more than 90% of cases.

Knee muscle strength was measured only 5 years postoperation. The isometric strength of the quadriceps and hamstrings of both knees was measured using a handheld dynamometer (HHD) [46,47], MicroFET3 (Hoggan Health Industries: West Jordan, UT, USA). The maximal force was expressed in Newtons (N). Participants were taught to perform isometric contractions of the knee muscles. They performed warm-up exercises for 5 min and two practice trials of the tests, rested for 30 s, and then performed the three measurement trials. Knee extension strength was measured according to the protocol described in other studies [46,47]. The HHD was positioned 2 cm proximal to the lateral malleolar tip with the knee at 60° of flexion and the hip at 90° of flexion. Knee flexion strength was measured with the participant placed on a stretcher in a prone position, with knees at 30° and hips at 0° of flexion. Arms were crossed under the participant’s forehead. The HHD was placed in the calcaneus at the level of the Achilles tendon insertion. Measurements were made three times for each limb by the same two researchers, and the average of the values was used for statistical analysis. If one of the intratest data points differed by ˃10% from the other data points, the measurement was repeated. During the tests, both in extension and flexion, participants were encouraged to make the maximum contraction.

The strength values used for the statistical assessment were as follows: Quadriceps Limb Symmetry Index (QLSI) = (injured quadriceps isometric peak/uninjured quadriceps isometric peak) ∗ 100
Hamstring Limb Symmetry Index (HLSI) = (injured hamstring isometric peak/uninjured hamstring peak) ∗ 100
H/Q ratio of the injured side = (injured hamstring isometric peak/injured quadriceps isometric peak)
H/Q ratio on the uninjured side = (uninjured hamstring isometric peak/uninjured quadriceps isometric peak).

For the controls, the nondominant leg was used as the injured leg, and the dominant leg was used as the uninjured leg.

### 2.3. Statistical Analysis

The statistical program used to assess results was R, version 4.0.2 (R Foundation for Statistical Computing: Vienna, Austria). The means, standard deviations, medians, and 25th and 75th percentiles were calculated for the quantitative variables. The Shapiro–Wilk test was used to check the normality of the data. Multiple linear regression for paired data was used to predict numerical variables as a function of time. The technique is a complete method to compare the evolution of numerical variables over time and in different groups. ANOVA was used to compare the means of the three groups since it is considered a very robust method despite the small sample size. Post hoc comparisons were made with the Tukey test for the regression and the ANOVA. A *p*-value of less than 0.05 was considered significant.

Using the R Studio “pwr” package, with α = 0.05 and power = 0.80, the sample size would be: For small effect size (0.1): 323 patients per group;For medium effect size (0.25): 53 patients per group;For large effect size (0.4): 22 patients per group.

Then, we reach the necessary sample size for a large effect size.

## 3. Results

Twenty-two patients in the ST group and 22 in the ST-G group were available for the assessment five years after ACLR. Two out of the initial twenty-six patients in the ST group underwent reoperation due to re-rupture of the reconstruction, and two were lost to follow-up. Regarding the initial 26 patients of the ST-G group, one patient underwent reoperation due to re-rupture, and three were lost to follow-up. The mean age at the time of the postoperative assessment was 31.7, 30.9, and 31.4 years for the ST, ST-G, and control groups, respectively (Table 1).

Table 2 presents the results of the pre- and postoperative side-to-side differences between the anterior tibial translation of the injured and the uninjured knees and the values of the pre- and postoperative PROMs. Side-to-side preoperative differences were >3 mm in both the ST and the ST-G groups and <1 mm in both groups postoperatively. These differences were highly significant between the pre- and postoperative periods but not significant when both operative groups were compared.

Minor but nonsignificant differences were observed preoperatively with the IKDC values of both groups. The median value for the ST-G group was 50.6 (47.6–57.7) preoperatively and 83.9 (77.5–90.5) postoperatively (*p* < 0.001). The median value for the ST group was 44.2 (35.6–55.1) preoperatively and 95.4 (90.8–97.7) postoperatively. The mean difference between the two groups at 5 years postoperation was 9.57 (CI 14.8–4.2) (*p* < 0.001). No significant differences between groups were detected both preoperatively and postoperatively for the Lysholm scoring. The preoperative vs. postoperative median values for the ST-G group were 58.5 (47.2–63) and 95 (94.2–100), respectively (*p* < 0.001). The preoperative vs. postoperative median values for the ST group were 52 (38.7–64.5) and 97.5 (95–99), respectively (*p* < 0.001).

Table 3 shows the values of the strength. Five years after the operation, the median QLSI was 102.5 (97–108.6) for the ST-G group, 96.7 (94.2–104.1) for the ST group, and 93.3 (87.9–99.6) for the control group. The mean differences were significant only between the ST-G group and the control group—9.62 (CI 17.4–1.7) (*p* = 0.01). The median HLSI was 88.3 (84–96.5) for the ST-G group, 95.2 (92–98.3) for the ST group, and 95.1 (91.6–97.3) for the control group. Differences between the ST-G and the other two groups were significant (*p* = 0.03), but there were no significant differences between the ST and control groups. The H/Q ratio was between 0.6 and 0.63 for all the groups. No significant differences were observed between groups, between the injured versus uninjured sides in the operated patients, or between the dominant versus nondominant knees in the controls.

## 4. Discussion

Several findings can be highlighted in this work. First, ATT differences between injured and uninjured knees were less than 1 mm after 5 years in both the ST-G and the ST groups, with no significant differences between them. These results are consistent with those in the current literature when the use of ST vs. ST-G grafts is compared [3,4,10,11,12,13], although these studies involve data with a maximum follow-up of 3 years. Since different techniques were used by these authors, it can be assumed that the use of grafts with one or two tendons shows no differences in ATT after ACLR.

The second finding to be highlighted in our study is that the ST-G group showed IKDC score results that were significantly lower than those of the ST group, while the Lysholm score showed no significant differences between the two groups 5 years postoperation. Although there is no consensus, scores between 85 and 90 are considered the threshold of success in PROM [18,21]. The IKDC values of our ST-G group did not achieve those considered within the normative values in a healthy knee population of 25 to 34 years old [48]. However, in the ST group, the values of the IKDC score were within the range of those considered normal for a healthy population of their age group [48]. The Lysholm knee score at 5 years was similar between the ST-G and ST groups, and the results were within the range of normative values for a population of this age [49]. These findings are consistent with those reported in the current literature [3,4,8,10,11,12,13].

In contrast to our findings, Sharma et al. [3], in a systematic review, concluded that the addition of gracilis harvest to an isolated semitendinosus harvest for ACLR showed no significant differences in patient-reported outcomes. More recently, others have found similar results, showing no significant differences in PROMs regarding the use of ST grafts alone or combined with gracilis harvest [8,10,12]. However, the mean follow-up of these studies was not longer than 3 years.

The third finding of our study to be highlighted is the differences observed in muscle strength 5 years after ACLR. There were no differences in the H/Q ratio in either the injured or uninjured knees of the ST and ST-G groups or between the two knees of the control group. No significant differences in the H/Q ratio of either knee were observed among the three groups. The values observed were within the range from 0.5 to 0.75, considered for an average population [39,50,51]. In contrast, a recent study reported fewer flexor strength deficits when just one tendon graft, compared to two tendon grafts, was used 36 months after ACLR [10].

Compared with healthy controls, patients with ACLR have a deficit in the activation of the quadriceps bilaterally [52,53,54], different movement patterns in the coronal and sagittal planes [32,53,55], and different volumes of the leg musculature [21,36]. There is a neural regulation to maintain symmetry of the extremities and an agonist–antagonist balance to dynamically stabilize the injured knee [39]. This compensatory mechanism includes facilitating hamstring and quadriceps inhibition [41]. The H/Q ratio may indicate a muscle imbalance around the knee joint [40,41]. The quadriceps and hamstring LSI are common methods of assessment success in restoring strength after ACLR [35]. The values of less than 90% should not be accepted as good results [18,56,57], although a more restrictive criterion of >95% is currently suggested [56]. Asymmetry in limb strength >15% after ACLR is a predictor of reinjury and decreased sports performance parameters [47]. In our study, the QLSI was >95% in both operated groups and 93.3% in the control group. However, HLSI for the ST and control groups was >95%, while the ST-G group did not even achieve the widely used 90% LSI cutoff value [32,35].

It has been shown that hamstring autograft harvesting can reduce knee flexor strength for up to 1–2 years after ACLR [58]. A decrease in relative hamstring strength combined with high relative quadriceps strength may present a potential risk factor for ACL tears [43]. Moreover, Monaco et al. [19] found that using the isolated ST tendon produced significantly better flexural force recovery than the full tibial tunnel technique with ST-G for ACLR. Sharma et al. [3] determined that the addition of gracilis to a single semitendinosus for ACL reconstruction results in statistically significant, but likely not clinically relevant, differences in hamstring strength. A recent study comparing outcomes of ST and ST-G tendon harvesting concluded that there was a significantly lower deficit in deep flexor strength, with better clinical outcomes at 3 years, in the group of only ST [10]. Kouloumentas et al. [12] concluded that all-inside ACL reconstruction with a short ST4 graft provides an advantage over ACL reconstruction with an ST/G graft in terms of improved knee flexion strength at higher angular velocities.

The results of this study suggest that the initial hypothesis should be rejected. Although the Lysholm score 5 years after the operation was not different between the two groups, the IKDC score was significantly lower in the ST-G group. The HLSI in this group 5 years postoperation was also significantly lower than that in the ST and control groups, with no significant differences between these two groups. Although no functional tests were carried out at follow-up in this study, the observed results let us suggest that the use of one autograft hamstring tendon for ACLR is clinically preferable to the use of two hamstring tendons. Further research with an appropriate number of patients is necessary to study how factors such as body mass index, type of sports activity, gender, and sports performance influence the differential results after using one vs. two autograft tendons.

This study has several limitations. First, we use HDD to measure force. The H/Q ratio is highly dependent on angular momentum, angular velocity, and contraction type [50]. This type of information can be obtained only by making measurements with an isokinetic dynamometer. Nevertheless, there is sufficient evidence today to consider that manual devices provide adequate knee muscle strength information with high reliability and reproducibility. They are very useful tools for standard clinical settings [46,47,59,60,61,62]. Second, we used the uninjured leg as a “healthy” comparison for the injured leg, despite a study reporting a decrease in the strength of both knees in unilateral ACL-deficient patients [63]. The influence of ACL injury on the strength of the uninjured knee would only be possible to determine if the preinjury values were obtained [64]. Therefore, it would be more appropriate to use normative values of strength adjusted for sex, body mass index, and activity level [63]. Third, all participants were males. We do not know if the results would be different in a group of unilateral ACL-deficient women. Fourth, we were able to measure strength only at 5 years of the study. Therefore, we do not know how knee muscle strength was before the operation or how it evolved from the first postoperative year. However, most authors comparing knee strength after ACLR with ST versus ST-G do not provide preoperative results [3,10,12,19,65]. We also do not know if a shorter follow-up time would produce results similar to those of other studies. Our results were based on PROMs, and we do not know if functional tests would have shown differences between patient groups. Although participation in recreational sports activity at follow-up was a condition to be included in the study, we do not know if there were differences in sports performance of the patients between the preinjury level and the final follow-up. Another important limitation is that the number of cases was small, and the study could be underpowered. The surgical technique was not the same in both groups, which could influence the results. However, the ATT was nearly the same for both groups postoperatively, and others have also reported that the technique used has little influence on the result’s differences when using one or two hamstring tendon grafts [3,7]. Finally, the clinical tools used could have some influence on our results. However, the KT1000 is the most used device to measure ATT [66], and the reliability of Microfet, the device used to measure isometric strength, is >0.95 [67,68].

## 5. Conclusions

IKDC scores were significantly lower in the ST-G graft group than in the ST group, but the Lysholm score showed no significant differences. The HLSI of the ST-G group was significantly lower than that of the ST and control groups, with no differences between these two. No significant differences in the H/Q ratio of the injured versus uninjured knees of the operated patients, nor the H/Q ratio of the nondominant versus the dominant knees of the controls, were observed 5 years postoperation. The QLSI was not significantly different between the three groups. No significant differences in ATT of reconstructed knees were detected between the ST and ST-G groups 5 years postoperation. This study shows that using just the ST autograft tendon for ACLR produces less flexor muscle strength decrease and better results on some PROMs than using both the ST and G autograft tendons.

## Figures and Tables

**Figure 1 jcm-11-05682-f001:**
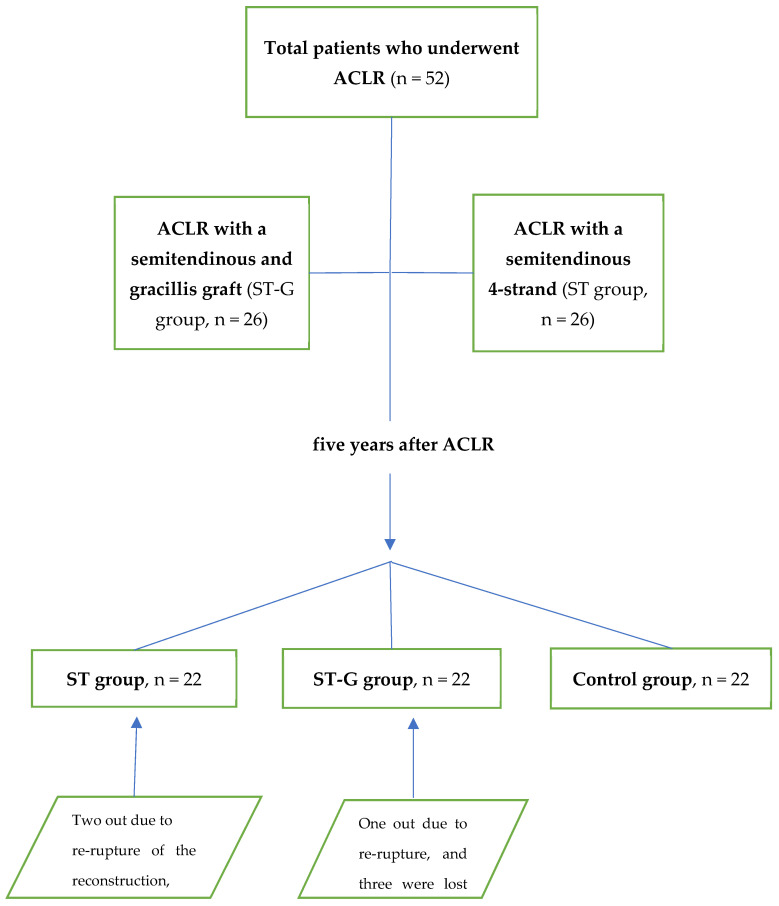
Method of assessment.

**Table 1 jcm-11-05682-t001:** Demographic data of the patients and controls. Preop—Preoperatively; ST—semitendinosus. ST-G—semitendinosus and gracilis. Data are means ± SD and median (25–75% IQR).

	Preop (n = 26 Each Group)	5 Years Postoperation (n = 22 Each Group)
	ST group	
Age	26.8 ± 8.5 25.5 (16–41)	31.7 ± 8.5 30.5 (21–46)
Sex (Male)	100%	100%
	ST-G group	
Age	25.8 ± 8.9 27 (16–49)	30.9 ± 8.8 32 (21–54)
Sex (Male)	100%	100%
	Controls	
Age	-	31.4 ± 8.6 30 (20–46)
Sex (Male)	-	100%

**Table 2 jcm-11-05682-t002:** Results of the two groups of patients (median and 25–75% interquartile range). STSD—Side-to-side anterior tibial translation differences (injured vs. uninjured knee) in mm; Preop—Preoperatively; ST—semitendinosus; ST-G—semitendinosus and gracilis. ^a^
*p* < 0.001 when comparing the preoperative and 5-year postoperative groups; ^b^
*p* < 0.001 when comparing the ST and ST-G groups.

	Preop	5 Years Postoperation
	STSD	
ST-G group	4 (2.2–5)	0.5 (0–1)
ST group	4 (3.5–4)	0.75 (0–1)
	IKDC	
ST-G group	50.6 (46.8–56)	83.9 (77.6–90.5) ^a^
ST group	44.25 (35.6–55.1)	95.4 (90.8–97.7) ^a,b^
	Lysholm	
ST-G group	58.5 (47.7–63.2)	95 (94.2–100) ^a^
ST group	52 (38.7–64.5)	97.5 (95–99) ^a^

**Table 3 jcm-11-05682-t003:** LSI and H/Q ratio strength 5 years after reconstruction and in the control group. Results are expressed as medians (25–75% interquartile range). QLSI—Quadriceps Limb Symmetry Index; HLSI—Hamstring Limb Symmetry Index; H/Q—Hamstring to Quadriceps Ratio; ST—semitendinosus; ST-G—semitendinosus and gracilis. The values of the injured side were compared with those of the nondominant side of controls. The values of the uninjured side were compared with the values of the dominant side of controls. ^a^
*p* =0.01 when comparing the ST-G and control groups. ^b^
*p* = 0.033 when comparing the ST-G with ST and control groups.

	QLSI	HLSI	H/Q Injured Side	H/Q Uninjured Side
ST-G group	102.5 (97–108.6) ^a^	88.3 (84–96.5) ^b^	0.6 (0.5–0.6)	0.63 (0.6–0.7)
ST group	96.7 (94.2–104.1)	95.2 (92–98.3)	0.61 (0.5–0.6)	0.63 (0.5–0.7)
Control	93.3 (87.9–99.6)	95.1 (91.6–97.3)	0.61 (0.4–0.7)	0.61 (0.4–0.7)

## Data Availability

The data that support the findings of this study are available from the corresponding author, G.L.G., upon reasonable request.

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
