# Peer review of "Isometric Knee Muscle Strength and Patient-Reported Measures Five Years after Anterior Cruciate Ligament Reconstruction: Comparison of Single versus Dual Autograft Hamstring Tendon Harvesting"

_jcm, 2022, doi:10.3390/jcm11195682_

Round 1

Reviewer 1 Report

Dear Authors, 

I have thoroughly reviewed your manuscript. The manuscript is well written, the results are interesting and clinically relevant. The discussion provides an adequate comparison with the current literature. The study limitations are well presented.

Author Response

Dear reviewer. Thank you very much for your revision and comments.

Reviewer 2 Report

This study aimed to study if, five years after the operation, the differential muscle strength between the reconstructed and its contralateral uninjured knee is different when the ST versus ST-G are harvested. There are several issues, which need to revised to improve clarity and quality of the manuscript.

Major comments:

1. A priori sample size was not estimated.

2. Why authors have followed up for a long time of 5 years without any mid-data between 1, 2 or 3 years?

3. Authors can add a figure to present study procedure.

4. Demographics of the study participants should be provided in Table.

5. One of the limitations identified by the authors that they used the uninjured leg as a “healthy” comparison for the injured leg. However, in the method section authors indicated that a third group of 22 non-injured 118 male controls was recruited. What was the purpose of this group?

6. Authors may include a few figures to present their findings  to improve clarity.

7. Authors may highlight scientific contribution of the current study and recommendation for future research direction. 

8. Limitations should be included in a single paragraph.

Author Response

Dear reviewer. Thank you very much for your revision and comments, which have been very useful. Please find enclosed our answers. We will be very grateful if you reviewed our manuscript again. We also thank your advice to improve the article.

Kind regards

Reviewer 3 Report

thanks for submitting this paper

the idea is good and the topic is interesting

there are some major methodological issues to be addressed

specific comments :

-please be sure that references are well formatted, relevant, and up to date

-please be sure about the English grammar

-I believe the whole manuscript needs somehow revision from a proficient scientific writer

-introduction is too long, no need to explain all your outcome measures, nor the surgical techniques. be focused on why is this study needed.

-is this prospective or retrospective? please specify

-number of participants is unclear, line 91 is 22, line 100 is 53 and so on.... please report the final number just once, preferably in the results section

-line 91 says "observational"... line 114 says "matched at the operation"... this study design is unclear

-line 118, this 3th group make no sense to me. I would remove this from the study. this is not even reported in your aims.

-line 151, how was reliability assessed

-there is no sample calculation, the study could very well be underpowered

-keep discussion focused

Author Response

Dear reviewer. Thank you very much for your revision, suggestions and comments, which have been very useful. We would be very grateful if you reviewed again our manuscript to be published in JCM. We also thank your advice to improve the article.

Kind regards

Reviewer 4 Report

1.      Please include all of the author’s emails after affiliation with name initials, except for the corresponding author based on MDPI format.

2.      In the abstract section, quantitative data must be included.

3.      At the end of your abstract, please provide a "take-home" message.

4.      Based on the MDPI format, all of the keywords must be written in lowercase.

5.      The Reviewer do not see the novel in the present article. My examination revealed that several similar previous publications appear to appropriately address the issues you have brought up in the current submission related to single/dual autograft comparison after follow up in several years. Please emphasize it more advance in the introduction section if there are any more truly something really new.

6.      Previous studies must be explained in the introductory part, including their work, innovation, and limits, to demonstrate the research gaps that will be filled in the current study.

7.      Line 82-83 “To the best of our knowledge”, it is not the scientific way, critical flaws. The authors needs to base on main databases such as Scopus, Web of Science, and PubMed. For Example, based on the literature searching in Scopus, Web of Science, and PubMed using keywords “Autograft” ……….

8.      In the last paragraph of the introduction, please explain the objective of the present article.

9.      Why the present study conducted as in vitro? in depth explanation is mandatory required. The discussion of potential in silico study should be included. The introductory and/or discussion part of an article should contain this important information. Also, the MDPI's suggested reverence should be applied in the explanation as follows: Jamari J, Ammarullah MI, Santoso G, Sugiharto S, Supriyono T, Heide E van der. In Silico Contact Pressure of Metal-on-Metal Total Hip Implant with Different Materials Subjected to Gait Loading. Metals (Basel). 2022;12(8):1241.

10.   To improve the reader's understanding of the materials and methods section simpler for them to grasp rather than only relying on the predominate text as it currently exists, the authors could incorporate figures that illustrate the workflow of the current study.

11.   What is the baseline of participant selection? Is there any protocol, standard, or basis that has been followed? It is unclear since the patient is very heterogeneous with a small number. The resonance involved impacts the present result makes this study flaws. One major reason for rejecting this paper.

12.   Basis of statistical analysis needs more exmpalantion.

13.   It is necessary to provide more information on the manufacturer, country, and specifications of the tools.

14.   Error and tolerance of clinical tools used in this work are important information that needs to be explained in the manuscript. It is would use as a valuable discussion due to different results in the further study by other researcher.

15.   Findings must be compared to similar past research.

16.   Before moving on to the conclusion section, the present study's limitation must be added at end of the discussion section.

17.   In the conclusion, please explain the further research.

18.   The reference should be enriched with literature from the last five years. Literature published by MDPI is strongly recommended.

19.   In the whole of the manuscript, the authors sometimes made a paragraph only consisting of one or two sentences that made the explanation not clearly understood. The authors need to extend their explanation to become a more comprehensive paragraph. In one paragraph, it is recommended to consist of at least 3 sentences with 1 sentence as the main sentence and the other sentences as supporting sentences. See line 138-141.

20.   Because of grammatical faults and linguistic style, the authors must proofread the document. MDPI English editing service would be a solution.

21.   Ensure that the authors followed the MDPI format exactly, edit the current form, and double-check all of the previously noted problems.

Author Response

(The authors gave the same response as above.)

Reviewer 5 Report

Thank you for the opportunity to review this article.
The article is a comparative observational study between ACL reconstruction with semitendinosus x 2 + gracilis x 2 tendons and with only semitendinosus x 4 tendon with all-inside technique.
The study is very interesting as it compares the functional results with a long follow-up of two of the most commonly used techniques.
I think the writing and methodology are impeccable. The introduction provides all the elements useful for understanding, the methods are complete and clear, the statistical analysis methodologically correct, the results clear, and the conclusions consistent with the results.
I would only suggest that the authors include a mention in the abstract to the use of the all-inside technique.
Thank you.

Author Response

(The authors gave the same response as above.)

Round 2

Reviewer 2 Report

Authors made all the required changes as recommended. No further comments.

Author Response

Thank you very much for your comments. Regards

Reviewer 3 Report

-as stated by the author's grammar must be revised again

-need at least post-hoc sample size calculation

-report reliability measures

Author Response

Thank you very much for your comments and suggestions. Please find enclosed our answers. Kind regards

Reviewer 4 Report

Reviewers greatly appreciate the efforts that have been made by the author to improve the quality of their articles after peer review. I reread the author's manuscript and further reviewed the changes made along with the responses from previous reviewers' comments. Unfortunately, the authors failed to make some of the substantial improvements they should have made making this article not of decent quality with biased, not cutting-edge updates on the research topic outlined. In addition, the author also failed to address the previous reviewer's comments, especially on comments number 5, 6, 7, 9, 11, 12, and 14.

Author Response

(The authors gave the same response as above.)
